# The Translation of Mobile-Exoneuromusculoskeleton-Assisted Wrist–Hand Poststroke Telerehabilitation from Laboratory to Clinical Service

**DOI:** 10.3390/bioengineering10080976

**Published:** 2023-08-18

**Authors:** Wanyi Qing, Ching-Yi Nam, Harvey Man-Hok Shum, Marko Ka-Leung Chan, King-Pong Yu, Serena Sin-Wah Ng, Bibo Yang, Xiaoling Hu

**Affiliations:** 1Department of Biomedical Engineering, The Hong Kong Polytechnic University, Hong Kong; 2Community Rehabilitation Service Support Centre, Queen Elizabeth Hospital, Hong Kong

**Keywords:** stroke, upper limb, robot, rehabilitation, telerehabilitation

## Abstract

Rehabilitation robots are helpful in poststroke telerehabilitation; however, their feasibility and rehabilitation effectiveness in clinical settings have not been sufficiently investigated. A non-randomized controlled trial was conducted to investigate the feasibility of translating a telerehabilitation program assisted by a mobile wrist/hand exoneuromusculoskeleton (WH-ENMS) into routine clinical services and to compare the rehabilitative effects achieved in the hospital-service-based group (*n* = 12, clinic group) with the laboratory-research-based group (*n* = 12, lab group). Both groups showed significant improvements (*p* ≤ 0.05) in clinical assessments of behavioral motor functions and in muscular coordination and kinematic evaluations after the training and at the 3-month follow-up, with the lab group demonstrating better motor gains than the clinic group (*p* ≤ 0.05). The results indicated that the WH-ENMS-assisted tele-program was feasible and effective for upper limb rehabilitation when integrated into routine practice, and the quality of patient–operator interactions physically and remotely affected the rehabilitative outcomes.

## 1. Introduction

Self-help home-based telerehabilitation after stroke with minimal professional assistance and remote supervision has a large demand to augment the traditional center-based and face-to-face outpatient service after the world suffered from the interruptions of the COVID-19 pandemic. Motor restoration of wrist–hand function after stroke is challenging and usually relies on outpatient services because of a delayed motor return in the distal function compared to the proximal joints in the upper limb rehabilitation in a short hospital stay constrained by the insufficiency of inpatient beds and professional manpower in public healthcare systems even in developed countries [1,2].

Compared to technologies, e.g., virtual reality games [3], a robot is more helpful in assisting repetitive and intensive physical training poststroke when professional manpower is insufficient because of the torque required to assist a paralyzed limb [4]. However, most robots adopted in the clinical service with rehabilitative effectiveness are large systems requiring professional operation in institutional environments [5]. Although some wrist–hand training systems, e.g., lightweight, soft robots, have been developed for potential home usage [6,7], few have been validated on their feasibility for self-help training at home with needed treatment effects by trial studies. One obstacle is the rehabilitative efficacy of the robotic designs targeting home usage by nonprofessionals when power and mechanical assistance are lowered for safety concerns [8], as most systems can neither effectively promote voluntary motor efforts (VMEs) from the paralyzed neuromuscular system for neuroplasticity with long-lasting effect, nor suppress proximal compensation, e.g., shoulder/elbow motions, in the upper limb, leading to learned disuse in the wrist–hand muscles [9]. Another obstacle lies in the preparation and management of telerehabilitation, whose translation into real clinical practice has not been studied yet. Challenges can occur when translating successful research trials into general clinical services, which might compromise the rehabilitative gains reported in the trials. For example, compliance with trial-successful protocols could be flexible in clinical services, even in face-to-face robot-assisted treatments [10], where therapists in the service were prone to involve their clinical habits in implementing routine practice rather than mechanically copying the research protocols. In addition, translating self-help telerehabilitation into service practice would meet more disturbances, such as the readiness of a patient to perform independent training at home and the experience of a therapist in remote supervision to provide timely support, besides the common concerns regarding safety issues and easiness of using the device discussed elsewhere [11].

In our pioneer work, a mobile exoneuromusculoskeleton (ENMS) was designed for poststroke upper limb rehabilitation [12], which is commercially available (Thecon Technology (HK) Limited, Hong Kong, P.R. China) (Figure 1a,b). Electromyography (EMG) signals were used in the system control to promote VMEs from the elbow, wrist, and hand for effective motor restoration [13]. By integrating neuromuscular electrical stimulation (NMES) and pneumatically actuated mechanical assistance, the system could achieve close-to-normal muscular coordination with suppressed compensatory motions in the whole upper limb and yield better motor outcomes and faster recovery than those solely using either mechanical assistance or NMES [14,15,16]. The ENMS’s rehabilitative effects were shown in a pilot trial on patients with chronic stroke when onsite assistance from a research professional was provided [12]. Moreover, a pilot trial was conducted using the wrist–hand module of the ENMS (WH-ENMS, Figure 1c) for self-help home-based telerehabilitation [17]. All participants in the study could use the system independently at home or with the help of their caregivers and completed the training according to the protocol with significant motor improvements in the upper limb without adverse or safety reports [17]. In this study, we translated the WH-ENMS-assisted self-help telerehabilitation to a local public rehabilitation center with the purpose of investigating its feasibility for routine practice and comparing the rehabilitative effectiveness between the research-based training and the practice in a real service by a non-randomized trial.

## 2. Materials and Methods

### 2.1. WH-ENMS and Preparation of Translation

The WH-ENMS provides pneumatic actuation with NMES driven by residual EMG from the paretic muscles of a stroke survivor, assisting (1) wrist extension with the hand open and (2) wrist flexion with the hand closed [17]. The WH-ENMS is lightweight (45 g for the wearable part on the wrist–hand) and easy to use, interfaced by a mobile application (App) on a smartphone, which communicates with the control box wirelessly via Bluetooth.

EMG-triggered control was adopted with a preset triggering threshold (i.e., three times the standard deviation (SD) above the baseline to indicate the VME) in the WH-ENMS [12]. Two muscle unions, the extensor carpi ulnaris (ECU) and extensor digitorum (ED), and the flexor carpi radialis (FCR) and flexor digitorum (FD) (i.e., ECU-ED and FCR-FD) were used as the voluntary neuromuscular drives to initiate the mechanical and NMES assistances from the WH-ENMS in each motion phase. Two channels of EMG-NMES were used to detect the EMG signals and deliver the NMES through two pairs of reusable surface electrodes placed on the common area of muscle bellies’ motor points of the ECU-ED and FCR-FD [17]. This electrode configuration was successful for the device control [12] and feasible for the self-help operation [17]. Once the system was initiated, the pneumatic fingers inflated to provide extension torque and deflated passively for voluntary finger flexion. The NMES assistance was used to assist wrist extension with the hand open via ECU-ED and wrist flexion with the hand closed via FCR-FD [12]. A reference electrode was attached to the skin surface of the olecranon to reduce the common-mode noise. Markers on the skin were used when teaching subjects how to position electrodes correctly.

The telerehabilitation program consisted of 20-session WH-ENMS-assisted upper limb training, with an intensity of 3–5 sessions/week, 1 session/day at most, and completed in 7 consecutive weeks, as detailed in [17]. The first 3 sessions and a prior tutorial (i.e., mandatory courses) were provided to persons after stroke and/or their caregivers under onsite professional assistance and supervision for preparation of the following self-help sessions at home. Remote and professional supervision of training progresses based on the automatic feedback from the WH-ENMS, e.g., start/end time of a training and repetitions of wrist–hand tasks, was provided in the home-based sessions. Our previous self-help telerehabilitation trial was a research study in which the mandatory courses were conducted in a laboratory environment by a full-time researcher who also supervised the home sessions remotely [17]. In this work, we translated the program to the Community Rehabilitation Service Support Centre (CRSSC) of the Hospital Authority of Hong Kong. A 2-hr introductory seminar with system demonstration by the research team was first delivered to the clinical team of CRSSC, composed of a registered occupational therapist (OT), an occupational therapy assistant (OTA), and a clinical engineer (CE) involved in the study, for briefing the telerehabilitation program and discussing the translational procedure. The OT and OTA were responsible for the direct interaction with patients, while the CE communicated with the device provider and the research team for system maintenance in case of need. The technical translation mainly included the skill of WH-ENMS configuration for individual patients, design and implementation of the WH-ENMS-assisted tasks for the upper limb, and supervision skills for home-based training through interactive tutorials between the teams within one week. Two WH-ENMS systems were adopted for parallel training of the routine practice by the OT and OTA within the regular opening hours of the CRSSC.

### 2.2. Home-Based Self-Help Telerehabilitation Program

The study was a non-randomized trial design with the CONSORT flow chart shown in Figure 2.

#### 2.2.1. Participant Recruitment

The human ethics approvals were acquired from the Human Participants Ethics Sub-Committee of the Hong Kong Polytechnic University (PolyU) and Research Ethics Committee (Kowloon Central and Kowloon East). Written informed consent was obtained from all participants before the program started.

The study was conducted in two different settings: (1) outpatients admitted to the CRSSC in Hong Kong were screened and recruited to attend the training in a clinic center setting (clinic group); (2) patients with a stroke from local communities were screened and recruited to attend the training in a neurorehabilitation lab setting (lab group) in the university. The inclusion criteria for both groups included the following: (1) single and unilateral brain lesion from a stroke that occurred longer than 12 months; (2) Modified Ashworth Scale (MAS) score less than 3 at the elbow, wrist, and finger [18]; (3) Fugl-Meyer Assessment of the upper extremity (FMA-UE) score over 15 [19]; (4) Mini-Mental State Examination score over 21 [20]; (5) detectable voluntary EMG signals of the driving muscles (ECU-ED, FCR-FD) on the paretic side (i.e., three times the SD above the EMG baseline); (6) Functional Independence Measure (FIM) of at least 51; and (7) fulfillment of the minimal requirements in the home, including a bridge chair without wheels, a table with a minimum surface area of 60 × 40 cm^2^ for the training space, and a 3G or above mobile network access. Patients were excluded if they (1) were epileptic, (2) had a cardiac pacemaker or other implants (e.g., neural implants), (3) had open wounds or skin lesions around the driving muscles, (4) had acute inflammation, (5) had shoulder pain, or poststroke central pain conditions, (6) had other neurological impairments besides stroke, or (7) were receiving other upper limb treatments during the telerehabilitation program period.

#### 2.2.2. Intervention Protocol

Both groups received the telerehabilitation program, assisted by the WH-ENMS. The program consisted of mandatory courses at the CRSSC or the lab, followed by self-help training sessions at home. The mandatory courses included a pre-training tutorial and three sessions of guided training (Figure 3a).

##### Pre-Training Tutorial

Prior to the training, each participant and their caregivers (if any) were provided with an introductory tutorial on donning and doffing the system, its operation, and the training protocols. Training parameters in the WH-ENMS for each participant, including the EMG threshold level, NMES assistance level, and mechanical assistance level, were set before the training and remained fixed throughout the 20 sessions for a participant, as detailed in [17]. If a participant did not have a smartphone, the experimental operators lent him/her one until they completed the program.

##### Training Protocol in Sessions

After the pre-training tutorial, 20 training sessions were provided to each participant. In each training session, participants were asked to sit by a table and maintain their shoulders above the table with a vertical distance of 30–40 cm. A smartphone with the App to provide visual clues during the training was placed on the table at a distance of 30–60 cm in front of the participant. Participants were required to perform repetitive limb tasks during the training. The repetitive limb tasks included a horizontal task (Figure 3d) and a vertical task (Figure 3e). The horizontal task referred to gripping a sponge (8.5 × 5.5 cm^2^) from the participant’s affected side, releasing it 50 cm laterally on the other side, and then returning it to the original place with target positions marked on the table. The vertical task referred to gripping a sponge under an 18 cm high shelf, releasing it at the top of the shelf, and then returning it to its original place. Each limb task was required to be repeated for 30 min. A ten-minute rest was allowed between the two consecutive tasks to prevent muscle fatigue. The three guided training sessions were supervised and assisted by the experimental operators at the CRSSC or the lab. During the guided training sessions, nearby professional assistance was provided at progressive levels, namely, (1) fully assisted level, where the experimental operators supported the participants from the training setup and supervised the entire training process in the first session; (2) semi-assisted level, where the participants completed the session mainly by themselves with minimum assistance from the experimental operators in the second session; and (3) independent-with-observation level, where the participants completed the training session independently under close observation by the experimental operators. An additional semi-assisted session was provided if the participants were not ready for the independent-with-observation session. If the experimental operators deemed a participant competent to perform the self-help training, they were required to conduct the remaining sessions at home. In the guided training sessions, onsite feedback by the operators was provided to enhance participants’ performance and to support their transition to self-help training at home. They were primarily prescriptive in nature, focusing on specific instructions and corrective suggestions to ensure competence in performing the training tasks. In the first self-help session, the experimental operators delivered the WH-ENMS with a charger and training props (i.e., a sponge and a shelf) to the participants’ homes, inspected the safety of the home, and observed a whole training session to ensure consistency with the guided sessions. In the self-help training sessions, operators’ feedback was tailored to solve technical issues encountered by participants, as well as to understand their overall training experiences. They were more descriptive in nature, aiming to provide explanations and guidance to help participants overcome challenges and improve their performance. The frequency and level of precision of the feedback varied depending on the specific needs and progress of each participant.

##### Logistics Management of Self-Help Training at Home

Training data, including the frequency of sessions, session duration, and complete movement cycles in a session, were recorded by the developed App and automatically transmitted to a server located in the neurorehabilitation lab at the university through a 3G or above mobile network after each session. Experimental operators in both groups remotely monitored the training data of each self-help session based on a prescribed training schedule agreed upon with the participant before the training. If the participant missed a session, the experimental operators contacted the participant by telephone or message to arrange a make-up session in accordance with the protocol training intensity. If a participant encountered any technical problems with the WH-ENMS at home, they were required to report it to the experimental operator immediately via telephone or message. A backup system was prepared for each participant before the training started and stored at the CRSSC or the lab for replacement. A malfunctioning system was replaced within a working day to avoid a violation of the training protocol. Additionally, the experimental operators contacted the ongoing participants weekly via phone or text message to discuss their experiences.

#### 2.2.3. Clinic Group versus Lab Group—Variations

Besides the common arrangement of the two groups stated above, there were variations, as summarized in Table 1. In the lab group, the participants received the telerehabilitation program assisted by the researcher (i.e., the experimental operator) with an academic background in neurorehabilitation engineering. The mandatory courses were conducted by the researcher in a training area within the neurorehabilitation lab at the university (Figure 3b). Participants started the training three days after the tutorial finished. During the guided sessions, the participants were required to avoid compensatory movements while performing the sponge transfer. Verbal corrections were provided immediately by the experimental operator once compensations were observed. For example, most stroke survivors were used to swaying the body trunk to compensate for the arm reaching of the affected side in the horizontal task, during which the patients were reminded to minimize the trunk motions. Additionally, hotlines and instant messaging channels were available to the participants for immediate communication 24 h every day during the self-help training period. There were no training fees charged for the telerehabilitation program in the lab group.

For the clinic group, the experimental operators were comprised of the registered OT and the OTA of the CRSSC. Participants in the clinic group attended mandatory courses in an independent training room at the CRSSC (Figure 3c). During the guided sessions and the first self-help session at home, the experimental operators in the clinic group supported and supervised participants’ training for about 30 min per session. This compromised arrangement deviated from the lab group because a therapist at the CRSSC needed to take care of several patients at the same time. The therapist could leave for another patient once the therapist evaluated that the participant could successfully perform the required limb tasks alone in that training session. Furthermore, the clinic group’s experimental operators were allowed to use different grasping objects in the training, such as a plastic apple or a plastic cup, to simulate the objects with different shapes in daily living. For the clinic group, the experimental operators were allowed to integrate rehabilitative elements adopted in the routine practice with the WH-ENMS-assisted upper limb training according to their personal experiences (e.g., the participants were allowed to complete the training tasks with compensatory movements) based on the pedagogy of the task-oriented rehabilitation, where compensation was also regarded as functional restoration once the task was achieved [21]. In addition, an optional add-on limb task, i.e., a 30 min forward task, was adopted by the operators for the participants in each training session for the clinic group, according to the center’s current practice in the upper limb rehabilitation, composed of gripping an object forwardly to a distance of 30 cm and returning it to the original place (Figure 3f). Thus, the duration of each training session in the clinic group ranged from 60 to 90 min. Each participant was required to pay HKD375 for the mandatory courses at the CRSSC, as per the routine management.

### 2.3. Evaluation of Training Outcomes

In the neurorehabilitation lab at the university, participants in both groups were evaluated by the clinical assessments before the pre-training tutorial (i.e., the pre-training evaluation), the day after the last training session (i.e., the post-training evaluation), and three months after the training (i.e., three-month follow-up evaluation, (3MFU)). The clinical assessments were conducted thrice within two weeks before the training as the baseline, with a minimum interval of two days between each assessment, to ensure the stability of the baseline before training. The mean of the three pre-training evaluations was used for the statistical calculations. EMG evaluations and kinematic evaluations were conducted at two time points (i.e., pre-training evaluation and post-training evaluation) to quantitatively measure the muscular coordination and kinematic performance of the paretic upper limb. The primary outcome of this study was the FMA-UE. The other clinical scores, EMG parameters, and kinematic parameters were considered secondary outcomes. In addition, a questionnaire was developed to evaluate the program’s usability and the participants’ motivation during the self-help training with remote support.

#### 2.3.1. Clinical Assessments

The clinical assessments for both groups were conducted by an assessor who was blinded to the protocol or grouping. Clinical measures included (1) the FMA-UE, which is a 66-score scale divided into 42 scores for the FMA of the shoulder and elbow (SE) and 24 scores for the FMA of the wrist and hand (WH). It is considered a reliable measure with wide applications to detect the motor function improvement of the upper limbs with robotic training [19,22]. (2) The Action Research Arm Test (ARAT), which has a total score calculated by the sum of 19 questions and evaluates the proximal and distal arm motor function [23]. (3) The FIM, which is an ordinal scale generally used to measure the disability degree in daily living [24]. (4) The Wolf Motor Function Test (WMFT), which consists of 17 tasks and records the time it takes to complete each task, is used to measure the motor ability of the upper limbs [25]. (5) MAS at the elbow, wrist, and finger flexors, which is the most widely used scale to assess muscle tone [18,26].

#### 2.3.2. EMG Evaluation

To quantitatively measure individual muscles’ activation and coordination, EMG signals from the ECU-ED, abductor pollicis brevis (APB), triceps brachii (TRI), biceps brachii (BIC), and FCR-FD muscles of the paretic upper extremities were recorded before and after the training. In an EMG evaluation session, maximum voluntary contractions (MVCs) of each muscle were first detected, which were followed by bare-arm testing trials of the horizontal and vertical tasks identical to those adopted as the training tasks with a repetition of 3 times, as detailed in [17]. A two-minute rest period was provided between two consecutive testing trials to avoid muscle fatigue.

The collected EMG signals were first amplified with a gain of 1000 (amplifier: INA 333, Texas Instruments Inc., Dallas, TX, USA), band-pass filtered from 10 to 500 Hz, and then sampled at 1000 Hz for digitization and stored for offline processing. Two EMG parameters were used to analyze the rehabilitative progress, i.e., the activation level of each target muscle and the EMG co-contraction index (CI) of a muscle pair [27].

The activation level of a muscle i was an averaged level with respect to its maximum value in MVCs and was obtained by first calculating,
(1)$$\bar{EMG}=\frac{1}{T}\int_{0}^{T}EMG_{i}\left(t\right)dt,$$
where $EMG_{i}\left(t\right)$ was the envelope of an EMG signal trial which was obtained by the rectification of the digitized EMG trial and then filtered by a 4th-order Butterworth low-pass filter with the cutoff at 10 Hz. Then, $\bar{EMG}$ was normalized by
(2)$$EMG_{i}=\frac{\bar{EMG}-{\bar{EMG}}_{rest}}{{\bar{EMG}}_{MVC}-{\bar{EMG}}_{rest}},$$
where $EMG_{i}$ was the activation level of muscle i, ranging from 0 to 1. ${\bar{EMG}}_{rest}$ was the EMG baseline level in a resting state. ${\bar{EMG}}_{MVC}$ was the largest value in MVCs. The EMG CI of a muscle pair evaluated the independence of the two muscles [27] and was calculated with every combination of ECU-ED, APB, TRI, BIC, and FCR-FD in this work,
(3)$$CI=\frac{1}{T}\int_{0}^{T}A_{ij}\left(t\right)dt,$$
where$\;A_{ij}\left(t\right)$ indicated the overlapping parts in the EMG signal envelope of muscle i and muscle j. T referred to the included time length. The larger the overlapping regions, the higher the CI value. Decreases in the EMG activation level of a muscle and the CI value of a muscle pair usually suggested released muscle tone and improved muscle coordination of the pair.

#### 2.3.3. Kinematic Evaluation

Kinematic measurements on motion smoothness and body trunk compensations were conducted by a motion capture system (Vicon Motion Systems, Oxford, UK) based on the standard marker configuration on the upper limb and body trunk [28].

The participants were required to perform the same bare-arm testing trials as in the EMG evaluation with a repetition of 3 times for each task. A break of 2 min between two consecutive trials was adopted to prevent fatigue. The number of movement units (NMUs) and maximal trunk displacement (MTD) were adopted to evaluate the motion smoothness and compensatory trunk movement. NMUs were the cumulated counts of signified change in the tangential velocity of the middle finger’s metacarpophalangeal joint in the testing trials [29], and an increase in NMUs indicated a decrease in movement smoothness. The MTD quantified the trunk displacements in the 3-dimensional space during the bare-arm testing trials with respect to an initial starting position [29]. The maximum value in any dimension of a testing trial was adopted as an MTD reading.

#### 2.3.4. Developed Questionnaire

A questionnaire was developed to assess the user’s experiences in the telerehabilitation program based on the Usefulness, Satisfaction, and Ease of Use Questionnaire (USE) [30] and the Intrinsic Motivation Inventory (IMI) [31]. USE was used in this study to assess the usability of the rehabilitation program assisted with the WH-ENMS with four dimensions: Usefulness, Ease of Use, Ease of Learning, and Satisfaction (Appendix A). USE has previously been applied to robot-assisted training [32] and employs a seven-point Likert scale, i.e., 1 = “strongly disagree” to 7 = “strongly agree”. The scoring of the USE was conducted for each dimension by summing up the item scores of the dimension and then dividing the sum by the total item score of that dimension to obtain the normalized score [33]. The original IMI is a multidimensional questionnaire that assesses participants’ experiences related to a target activity with 45 items in total and can be modified to fit specific activities based on different study designs [34]. In this study, a customized IMI questionnaire consisting of 28 items was adopted to measure the participants’ intrinsic motivation regarding the telerehabilitation program, divided into seven subscales: interest/enjoyment, perceived competence, effort/importance, pressure/tension, perceived choice, value/usefulness, and relatedness (Appendix A). A seven-point Likert scale (i.e., 1 = “not at all true” to 7 = “very true”) was used to rate each item. Subscale scores of the IMI were calculated by averaging all of the items on each subscale [31] and then normalizing them by the maximum response (i.e., 7).

#### 2.3.5. Statistical Analysis

The statistical analyses were performed by SPSS version 26 (IBM, Chicago, IL, USA). All outcomes were subjected to normality tests using the Shapiro–Wilk test [35]. After the Shapiro–Wilk test, the following parameters were confirmed to be normally distributed (*p* > 0.05): clinical scores of FMA-SE, FMA-WH, ARAT, WMFT score, and WMFT time in the lab group; EMG activation level of ECU-ED, TRI, and FCR-FD muscles in the lab group, BIC muscle in both groups; CI values of the ECU-ED/APB, ECU-ED/FCR-FD, ECU-ED/BIC, ECU-ED/TRI, FCR-FD/TRI, APB/TRI, BIC/TRI muscle pairs in the lab group, FCR-FD/APB muscle pair in the clinic group, FCR-FD/BIC and APB/BIC muscle pairs in both groups; NMUs in the lab group, MTD in both groups; USE and IMI scores in both groups.

For clinical outcomes, a one-way repeated measures analysis of variance (ANOVA) with a Bonferroni post hoc test was adopted to compare the scores before, after, and three months after the training within a group with normality satisfied; otherwise, a Friedman test with a Wilcoxon signed rank post hoc test was performed. The EMG and kinematic data were compared using a paired *t*-test, or Wilcoxon signed rank test, depending on the normality of the parameters for intragroup comparison. Quade’s ANCOVA was used to detect the difference at post-training evaluation and 3MFU with the mean of the three baseline scores as the covariate in the intergroup comparison of clinical scores. An independent *t*-test was used to compare the changes between the two groups after the training regarding the EMG parameters, kinematic parameters, USE, and IMI scores; a Mann–Whitney U test was used as a substitute for an independent *t*-test if normality was unsatisfied. Statistical significance was indicated by *p* values of 0.05, 0.01, and 0.001 in this study.

## 3. Results

In the period from April 2020 to December 2022, 19 outpatients admitted to the CRSSC were screened, and 13 were recruited in the clinic group. Meanwhile, 17 poststroke persons from local communities were screened, and 12 of them were recruited in the lab group. A total of 24 participants completed the telerehabilitation program in both groups. One recruited subject in the clinic group terminated the training after five sessions because of personal reasons. Demographic data of both groups are shown in Table 2. There were no statistical differences between the two groups in terms of age, gender, hemiplegic side, and stroke type (*p* > 0.05), except that the participants in the lab group were more chronic poststroke than the clinic group (*p* ≤ 0.05).

### 3.1. Behavioral Improvements in Clinical Assessments

There was no significant difference in the pre-training evaluation between the two groups in clinical assessments (*p* > 0.05; Table A1), except for a significant intergroup difference in the FIM (*p* ≤ 0.05). Figure 4 shows the measured clinical scores for each group across the pre- and post-training evaluations and 3MFU. Significant increases were obtained in the FMA-UE, FMA-SE, and FMA-WH for both groups after the training (*p* ≤ 0.05). The increments in the FMA-UE and FMA-SE for both groups were maintained at 3MFU (*p* ≤ 0.05), whereas the significant increase in the FMA-WH was only maintained in the lab group at 3MFU (*p* ≤ 0.05). Additionally, the lab group achieved significantly higher FMA-UE at 3MFU and FMA-WH at the post-training and 3MFU assessments than the clinic group (*p* ≤ 0.05). There was a significant increase for both groups in the ARAT after the training (*p* ≤ 0.05); however, the improved ARAT was only maintained by the lab group at 3MFU (*p* ≤ 0.05). Both groups had significant increases in the WMFT score after the training and maintained these improvements at 3MFU (*p* ≤ 0.05). However, only the lab group demonstrated a significant decrease in the WMFT time at the post-training and 3MFU assessments (*p* ≤ 0.05). The WMFT time for the lab group was significantly lower than the clinic group after the training (*p* ≤ 0.05). The decreases in the MAS scores at the elbow, wrist, and finger joints were statistically significant for both groups after the training (*p* ≤ 0.05), and the reduced MAS scores for the wrist and finger joints were retained for three months (*p* ≤ 0.05). The results of the inter- and intra-group comparisons were summarized in Table A2.

### 3.2. Improvements in Muscular Coordination by EMG

Multi-channel EMG signals were captured from muscles in the paretic upper limb in evaluation tasks of the two groups before and after the training. Figure 5 illustrates the measured EMG parameters (i.e., normalized EMG activation level and normalized EMG CI) before and after the training. The clinic group had a significant reduction in the EMG activation level of the ECU-ED muscle after the training (*p* ≤ 0.05), and the lab group obtained a significant reduction in the EMG activation level of the APB and FCR-FD muscles after the training (*p* ≤ 0.05). There were significant decreases in CI values in the ECU-ED/FCR-FD, ECU-ED/BIC, FCR-FD/APB, FCR-FD/BIC, FCR-FD/TRI, APB/BIC, and BIC/TRI muscle pairs in the lab group after the training (*p* ≤ 0.05), while the clinic group had a significantly lower CI value in the FCR-FD/APB muscle pair after the training (*p* ≤ 0.05). Moreover, the lab group achieved significantly lower values in the CI values of the ECU-ED/FCR-FD, ECU-ED/BIC, FCR-FD/APB, FCR-FD/BIC, and APB/BIC muscle pairs than the clinic group (*p* ≤ 0.05). The results of the inter- and intra-group comparisons of the EMG parameters were summarized in Table A3.

### 3.3. Improvements in Kinematic Performance

Three-dimensional motions of the paretic upper limb were captured in the evaluation tasks of the two groups before and after the training. Figure 6 shows the measured kinematic parameters (i.e., the NMUs and MTD) before and after the training. There was a significant decrease in the NMUs for both groups after the training (*p* ≤ 0.05). The MTD was decreased significantly in the lab group after the training (*p* ≤ 0.05), whereas the MTD was increased in the clinic group with no statistical significance (*p* > 0.05). There was a significant difference in the MTD between the two groups after the training (*p* ≤ 0.05). The results of the inter- and intra-group comparisons based on the pre- and post-measurements of kinematic parameters were summarized in Table A4. Comparing the amount of change in the NUMs and MTD after the training by independent *t*-tests between the two groups, the lab group (mean = −20.0958; 95% confidence interval from −38.7219 to −1.4698) demonstrated a more significant decrease in the MTD (*p* = 0.014) than the clinic group (mean= 10.9202; 95% confidence interval from −5.8523 to 27.6926). No significant difference was observed in the NUMs (*p* = 0.742).

### 3.4. Remote Monitoring of Training Progresses

For the clinic group, seven participants received the assistance of device operation from caregivers, while five participants completed the training at home independently. In the lab group, six participants received assistance with device operation from caregivers, and the others completed the training at home independently. The recorded logistic data are shown in Table 3. The average training frequencies of the participants were 3.33 ± 0.47 (mean ± SD) and 3.75 ± 0.72 (mean ± SD) sessions per week for the clinic and lab groups, respectively. The average training durations per session were 91.30 ± 8.50 (mean ± SD) minutes per session for the clinic group and 62.80 ± 1.93 (mean ± SD) minutes per session for the lab group. The average cycles of the completed movement were 184.23 ± 30.46 (mean ± SD) and 115.20 ± 9.50 (mean ± SD) cycles per session in the clinic and lab groups, respectively.

### 3.5. Questionnaire Outcomes

Eleven participants in the clinic group and ten in the lab group completed the questionnaire. Figure 7 illustrates the comparison of the normalized scores of the USE and IMI between the two groups. All the mean% of USE were over normalized neutral scores (50%), and the lab group had a higher mean% in each dimension of USE than the clinic group. For the IMI, most of the mean% in the lab group were higher than those of the clinic group. There was a significant difference in scores of IMI’s value/usefulness subscale (*p* ≤ 0.05), in which participants in the lab group considered the training to be more valuable than those in the clinic group. USE and IMI of each group after the training were summarized in Table A5.

## 4. Discussion

The results of this study support that it is feasible to translate the telerehabilitation program into a routine clinical service, and the motor recovery of the paretic upper limb was observed through clinical scores, EMG parameters, and kinematic parameters after the training.

The telerehabilitation program in the clinical service could improve the entire paretic limb’s voluntary motor function, which was manifested by the FMA-UE, FMA subscales (FMA-SE and FMA-WH), and ARAT (Figure 4). This finding was consistent with other studies on robot-assisted training of the distal joints after a stroke, where the upper limb’s proximal and distal joints obtained significant motor recovery [36,37]. Possible reasons for this included the following: (1) coordinated voluntary movements of the proximal and distal upper limb were required when performing repetitive limb tasks during the training [17]; (2) the proximal joints’ voluntary movements compensated for the movements of the distal joints during the training [37]; and (3) competition between the proximal and distal joints during the training involved both related muscles [38]. Moreover, the lab group had a significant improvement in the entire paretic limb’s voluntary motor performance, as indicated by the FMA-UE and ARAT (Figure 4). However, participants in the clinic group achieved less improvement in the distal joints’ voluntary motor function after the training and at 3MFU than the lab group, with no continuous effects for three months in the distal joints’ voluntary motor function (Figure 4). The different performances in the two groups’ clinical assessments might be related to the varied arrangement in the clinical translation, which was discussed later.

The WMFT is a measurement that assesses specific functional abilities of activities of daily life (ADLs) depending on the affected upper limb [25]. The increased WMFT score after the training and at 3MFU indicated an improved ability to participate in actions close to daily activities for both groups (Figure 4). The speed in task completion was also improved in the lab group, as represented by the increased efficient coordination indicated by the WMFT time (Figure 4). There was no significant change in the FIM after the training because it is commonly adopted to evaluate the independence in ADL performance across all aspects [24]. Furthermore, it also suggests that the WH-ENMS-assisted telerehabilitation did not benefit the overall ADL improvements for the recruited persons with chronic stroke.

In the clinic group, spasticity in the elbow, wrist, and finger was reduced significantly after the training, as indicated by the MAS scores (Figure 4). Similarly, the lab group had significantly decreased spasticity in the elbow, wrist, and finger after the training. Decreased spasticity can improve muscle coordination and joint stability [39], and the decrease continued for three months in the spasticity at the distal joints for both groups (Figure 4). However, a greater rebound in the tendency for spasticity at the distal joints was observed in the clinic group at 3MFU compared with the lab group (Figure 4). Therefore, the results suggested that the robot-assisted training in the lab group achieved a greater improvement in releasing muscle spasticity than the robot-assisted training in the clinic group.

The EMG activation level can reflect the neural control of a specific muscle [27]. A decreased EMG activation level of the target muscles suggested reduced spasticity and voluntary control with less motor effort to complete repetitive limb tasks. In the clinic group, the decreased EMG activation level of the ECU-ED muscle indicated a reduction in the excessive muscular activities at the distal joints, which was consistent with the results of the MAS scores at the distal joints (Figure 4 and Figure 5a). This implied that the decrease in MAS scores at the distal joints was related to the improved muscle control of ECU-ED. In the lab group, a significant reduction in the EMG activation level of the APB and FCR-FD muscles also suggested released spasticity at the distal joints, which implied that the decrease of the MAS scores at the distal joints in the lab group was related to the improved muscle control of the APB and FCR-FD (Figure 4 and Figure 5a).

Muscular coordination between muscle pairs was quantified by an EMG CI to indicate the co-activating patterns. A significant reduction in the CI values after the training suggested an overall improvement in independency between the muscle pairs. The decreased CI values in the FCR-FD/APB muscle pair for both groups implied fractionated joint control of the fingers and thumbs in hand grasping and releasing motions since the APB is the key muscle for digital opposition [40]. There was a release from the elbow compensation during the finger motions, which was indicated by a reduction of finger muscles/BIC CI values in the lab group. Excessive co-contractions are energetically expensive, and abnormal muscle co-activating patterns in the paretic limb after a stroke affect limb movements’ accuracy and efficiency [41]. For the lab group, the greater reduction in the CI values of the ECU-ED/FCR-FD, ECU-ED/BIC, FCR-FD/APB, FCR-FD/BIC, and APB/BIC muscle pairs indicated that fewer co-activating patterns of muscle pairs contributed to greater efficiency in skillful tasks and motor restoration for three months, which might explain the improved WMFT time after training and at 3MFU, and the continued effects in the FMA-WH and ARAT at 3MFU (Figure 4 and Figure 5b).

In the clinic group, the decreased NMUs after the training demonstrated smoother cross-joint movements of participants, which indicated enhanced fine motor control and improved joint coordination [29,42] (Figure 6a). Compared with the clinic group, the NMUs with a greater reduction in the lab group after the training might be related to improved muscular coordination demonstrated in the EMG CI patterns (Figure 5b and Figure 6a). Trunk compensation was identified by maximal trunk displacement during bare-arm testing trials (i.e., MTD). Participants adopted more trunk compensation in the clinic group, yet less in the lab group after the training, as indicated by the MTD (Figure 6b). In the lab group, the decreased trunk compensation might be due to more independent upper limb motions with the release of spasticity and improved muscular coordination for independent contraction. Although reduced spasticity and decreased EMG CI values in the paretic upper limb were also obtained in the clinic group, the poorer performance in the MTD of the clinic group is related to the different supporting schemes adopted in the clinical translation, such as how to complete the training tasks with the assistance of the same robot. This difference also resulted in the varied improvements between groups evaluated by the clinical scores and EMG parameters. In this work, the lab group was more chronic after a stroke than the clinic group. However, the lab group achieved better holistic motor improvements than the clinic group. It suggested that the time after stroke for the participants in this study was not a key determinant related to the rehabilitative outcomes.

In a practical clinical service, supporting schemes on how to complete a motion task relying on compensatory strategies have been widely adopted by therapists to guide stroke survivors in establishing their independence in ADLs in the early stages after a stroke for a quick discharge from the hospital, mainly related to a shortage of healthcare resources for hospital stays [43,44]. However, these compensatory strategies adopted in early rehabilitation limit further motor recovery in the chronic stage, such as distal joint functions, because of the learned disuse in the distal muscles [9,45]. In the clinic group, the experimental operators encouraged participants to complete limb tasks regardless of the extent of compensatory strategies. In the lab group, only necessary support was allowed to achieve minimized compensatory motions for effective motor restoration. The supporting scheme in the clinic group resulted in the adoption of compensatory strategies from the trunk and the unaffected upper limb, thereby limiting VMEs from the affected upper limb, which could be observed from the immediate training effects, such as less reduction of the EMG CI after training. Furthermore, motor patterns (i.e., compensatory strategies), once learned, are generalized to the participants’ ADLs [46]. Long-term maintenance of motor gains was also compromised in the clinic group due to the learned nonuse, as indicated by the rebound in the FMA-WH and ARAT at 3MFU in the clinic group (Figure 4). The additional training intensity in the clinic group (90 min/session) did not induce equivalent motor gains, compared with those for the lab group with only 60 min/session. The final rehabilitative effectiveness obtained in this study suggests that not only the intensity of robotic assistance but the supporting schemes adopted in training affect the motor relearning outcome. In this study, the rehabilitative gain demonstrated by the lab group (i.e., a well-controlled research trial) was compromised after the clinical translation because of the deviated supporting scheme in the clinical group.

The quality of patient–operator interaction, dependent upon interactive duration, promptness of feedback, and professional guidance, played a crucial role in the training outcomes [47]. Unfamiliarity, and even psychological resistance to new techniques, such as using robots and telerehabilitation settings, is a common challenge faced by neurologically impaired patients whose confidence and willingness to accept new things are much lower than the unimpaired [48]. High-quality patient–operator interactions could help to overcome the challenge by familiarizing patients with robot-assisted training, building their confidence, and increasing their involvement in self-help telerehabilitation, which ultimately contributes to better training outcomes. Moreover, patients’ trust in the professionality of the operator who introduced the new treatment to them secured the smoothness of the training later. In the clinic setting, the quality of the patient–operator interaction was constrained by the limited interactive time and delayed feedback. The average period supervised by the therapist was only 30 min per mandatory session. This was an average interactive period affordable in the routine practice of a professional at the CRSSC, a public rehabilitation center in Hong Kong, due to the shortage of resources [1]. In contrast, the operators in the lab group were able to provide one-on-one supervision throughout the entire training session for 60 min. Due to the limited onsite interactions, participants in the clinic group did not receive adequate guidance and prompt feedback, especially during the early sessions. In addition to onsite interactions, another challenge was timely feedback on inquiries during the self-help home-based sessions. The busy schedule of the public clinical services made it difficult for the therapist to provide timely feedback in the clinic group. The lack of timely support and feedback in ongoing sessions at home might negatively impact patient engagement and motivation. During the training in the clinic group, participants felt less supported and motivated to achieve motor restoration, which could be reflected by the lower scores of IMI compared with the lab group (Figure 7). As revealed by the USE results (Figure 7), participants in both groups reported holistic positive experiences with the usability of the WH-ENMS device (>50% normalized scores in all items). There was no difference between the groups on USE, although the clinic group suffered from fewer patient–operator interactions than the lab group. However, IMI results indicated that the clinic group showed less confidence or motivation in the training program than the lab group when professional support, or timely feedback, was reduced; this finally led to a significant group difference in perceiving the value/usefulness of the training program. Therefore, continuous support with timely feedback, whether onsite or remote, during the training could enhance the participants’ involvement and improve the overall outcome of the rehabilitation program.

The quality of patient–operator interactions can be elevated by improving the operator’s efficiency in supervision to deliver the necessary skills to the patient, or the caregiver, for the preparation of the relatively independent self-help sessions. The operator in the lab group in this study had more experience in the supervision of telerehabilitation by the WH-ENMS than that in the clinical group, who did not have prior experience in poststroke telerehabilitation [17]. This was demonstrated by the different usability perceived by participants in the two groups (Figure 7). In the clinic group, three participants considered the WH-ENMS difficult to learn and use as they scored less than 50% in every dimension of the USE. This indicated that the effective transfer of self-help skills in mandatory sessions could improve the usability of the new device. The supervising skill of the operator could be improved by technical training before service delivery and communication with experienced clinical researchers on techniques related to the rehabilitation mechanism, such as minimizing compensatory motions in the training and the self-accumulated experience in the service after serving more patients. Although statistical significance was achieved in the intra- and inter-group comparisons of this work, large-scale and multi-center clinical trials with larger sample sizes will be carried out in future works to further validate the efficacy of the ENMS-assisted telerehabilitation poststroke. Additionally, mental fatigue and movement quality monitoring in the training will also be investigated.

## 5. Conclusions

This study suggested that it was feasible to translate a self-help telerehabilitation program with WH-ENMS assistance from the lab to a practical clinical service with significant motor improvements. In this study, although the clinical group obtained less motor improvement compared with the lab group, the variation was mainly related to the less qualified patient–operator interactions, the varied compensatory supporting strategies adopted in the training, and the lower perceived usability by patients in the self-help training, which could be improved by the operator’s efficiency and accumulated experience in supervision to deliver the necessary skills to the patient or the caregiver, for the preparation of the relatively independent self-help session.

## Figures and Tables

**Figure 1 bioengineering-10-00976-f001:**
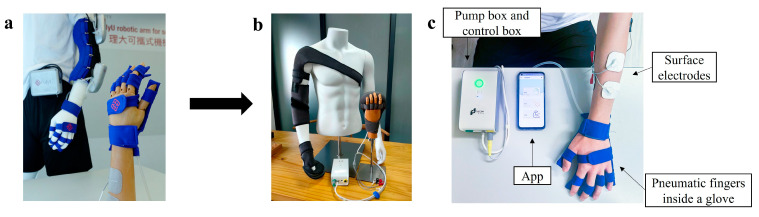
Evolution of ENMS from laboratory development to market availability. (**a**) The laboratory prototype of the ENMS developed in the lab; (**b**) the commercially available ENMS; (**c**) the wrist–hand module of ENMS for self-help telerehabilitation.

**Figure 2 bioengineering-10-00976-f002:**
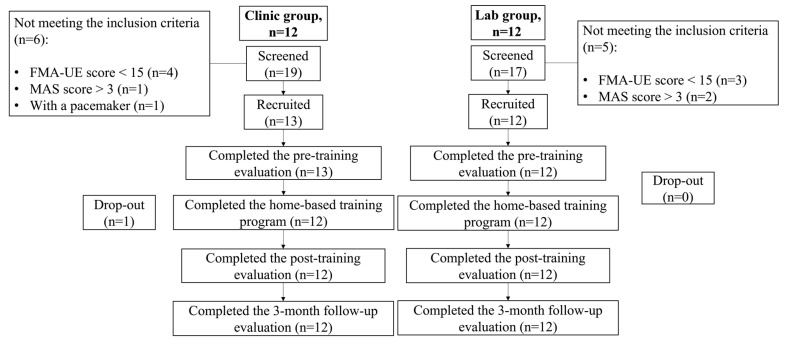
The Consolidated Standards of Reporting Trials flowchart of the study.

**Figure 3 bioengineering-10-00976-f003:**
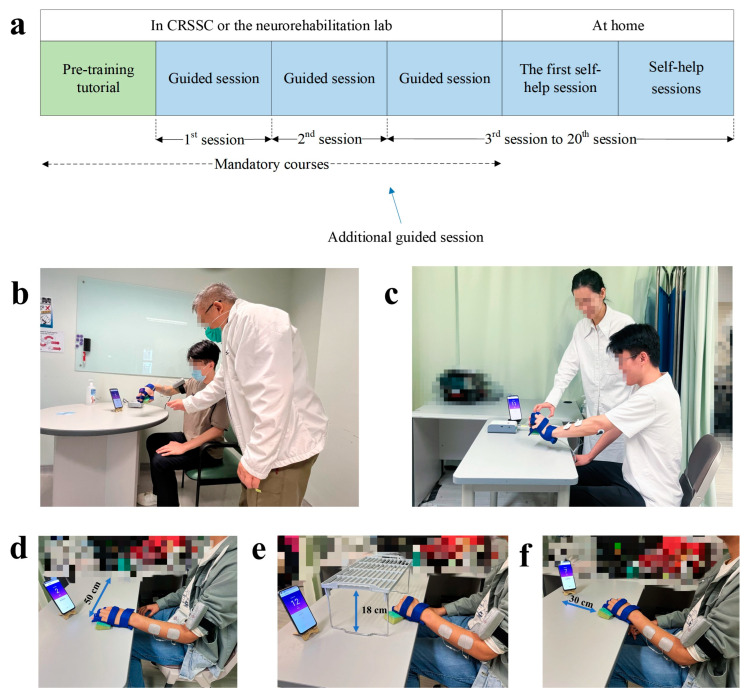
Timeline and training setup of the 20-session training program. (**a**) Timeline of the 20-session training program; (**b**) guided sessions assisted by an OT or OTA at the CRSSC; (**c**) guided sessions assisted by a researcher in the lab. During each training session, participants were required to perform repetitive limb tasks, including (**d**) a 30 min horizontal task and (**e**) a 30 min vertical task in both groups, and (**f**) an optional 30 min forward task at the CRSSC.

**Figure 4 bioengineering-10-00976-f004:**
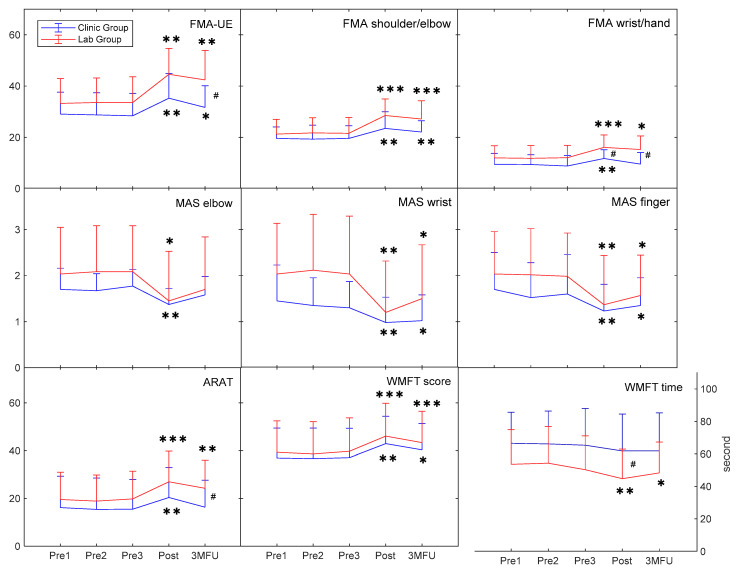
Behavioral improvements. Clinical scores by group before, after, and 3 months after the training represented by means and SDs. Significant levels are indicated by * (*p* ≤ 0.05), ** (*p* ≤ 0.01), and *** (*p* ≤ 0.001) for one-way repeated measures ANOVA intragroup tests or Friedman intragroup tests; # (*p* ≤ 0.05) for Quade’s ANCOVA intergroup tests on the time point with the mean value of three pre-training evaluations as the covariate.

**Figure 5 bioengineering-10-00976-f005:**
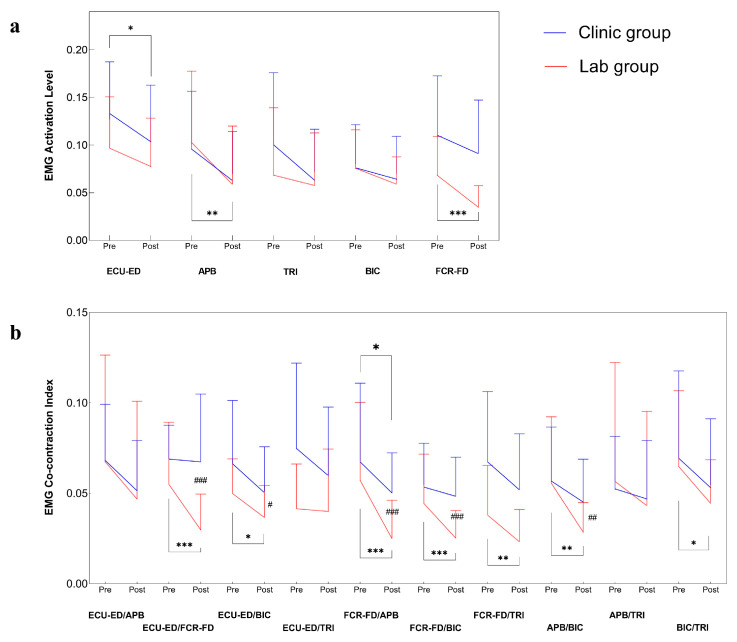
Improved muscular coordination. (**a**) Normalized EMG activation levels and (**b**) normalized co-contraction index by group before and after the training represented by means and SDs. Significant levels are indicated by * (*p* ≤ 0.05), ** (*p* ≤ 0.01), and *** (*p* ≤ 0.001) for paired *t*-test or Wilcoxon signed rank intragroup tests; # (*p* ≤ 0.05), ## (*p* ≤ 0.01), and ### (*p* ≤ 0.001) for independent *t*-test or Mann–Whitney U intergroup tests.

**Figure 6 bioengineering-10-00976-f006:**
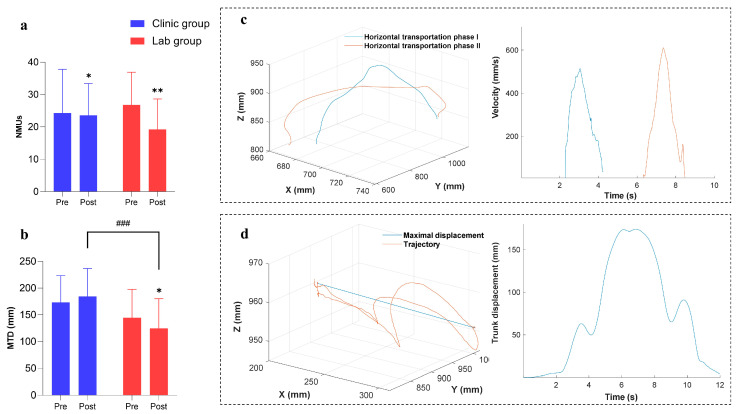
Improved kinematic performance. (**a**) NMUs and (**b**) MTD by group before and after the training represented by means and SDs. Significant levels are indicated by * (*p* ≤ 0.05), ** (*p* ≤ 0.01) for paired *t*-test or Wilcoxon signed rank intragroup tests; ### (*p* ≤ 0.001) for intergroup independent *t*-test. (**c**) Representative measured trajectory of the hand marker during the transport phases in the horizontal task for a participant and the related velocity profiles of the trial. (**d**) Representative measured trajectory of the thorax marker over the entire trial for the horizontal task for a participant and the related displacement profiles in the trial.

**Figure 7 bioengineering-10-00976-f007:**
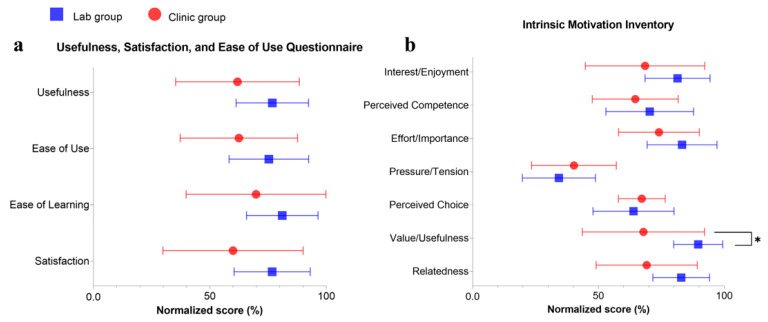
Normalized scores of (**a**) USE and (**b**) IMI by group represented by means (mean%) and SD (SD%). Significant levels are indicated by * (*p* ≤ 0.05) for intergroup independent *t*-test.

**Table 1 bioengineering-10-00976-t001:** Arrangement of trial implementation for the clinic and lab groups.

	Clinic Group	Lab Group
**Participants source**	Outpatients referred by rehabilitation doctors	Volunteers from local communities
**Evaluation**		
Venue	Neurorehabilitation lab at PolyU	
Assessor	The same blinded assessor	
**Mandatory courses**		
Venue	A treatment room, CRSSC	Neurorehabilitation lab at PolyU
Operator	Registered OT, OTA	Research staff
Supervision duration	The first 30 min/session	60 min
**Self-help training**		
Venue	Participants’ homes	
Training frequency	3–5 sessions/week	
Session duration	60–90 min/session	60 min/session
Remote training supervisor and contact	Registered OT	Research staff
Remote availability	9 am to 6 pm, Monday to Friday	Flexible whenever needed
**Withdrawal**	Yes, at any time point in the program	
**System Maintenance**	Referred by the CE to technicians of the research team	Technicians of the research team
**Charge to patient**	HKD375 at the CRSSC	Free

**Table 2 bioengineering-10-00976-t002:** Demographic characteristics of participants by group. The significant difference is indicated by * (*p* ≤ 0.05).

Characteristics	Clinic Group (*n* = 12)	Lab Group (*n* = 12)	*p*
Age ^a^ in years (mean ± SD)	53.33 ± 11.47	58.42 ± 13.47	0.203
Time since stroke ^a^ in years (mean ± SD)	3.32 ± 2.22	12.42 ± 10.88	0.003 *
Gender ^b^ (male/female)	7/5	6/6	0.682
Hemiplegic side ^c^ (left/right)	6/6	9/3	0.400
Stroke type ^c^ (ischemic/hemorrhagic)	3/9	6/6	0.400

^a^ Mann–Whitney U test. ^b^ Pearson Chi-square test. ^c^ Fisher’s exact test.

**Table 3 bioengineering-10-00976-t003:** Logistic data of participants by group. The means and SDs for each parameter of logistic data.

Parameters	Clinic Group (*n* = 12)	Lab Group (*n* = 12)
Frequency (session/week)	3.33 ± 0.47	3.75 ± 0.72
Duration (min/session)	91.30 ± 8.50	62.80 ± 1.93
Complete movement cycles (cycle/session)	184.23 ± 30.46	115.20 ± 9.50

## Data Availability

All data from this study are available from the authors upon reasonable request.

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
