# Peer review of "The Translation of Mobile-Exoneuromusculoskeleton-Assisted Wrist–Hand Poststroke Telerehabilitation from Laboratory to Clinical Service"

_bioengineering, 2023, doi:10.3390/bioengineering10080976_

Round 1
Reviewer 1 Report
1. Line 83: While the rationale of tenodesis principle is to perform wrist extension to assist hand grasping, the authors should explain why they designed WH-ENMS by assisting (1) wrist extension with the hand open and (2) wrist flexion with the hand closed, which is contradictory to the tenodesis principle.
2. Line 90: Why is WH-ENMS designed to detect EMG from extensor carpi ulnaris instead of extensor carpi radialis? Extensor carpi radialis is apparently a larger muscle and easier to locate.
3. Line 155: Since the outcome measures of this study were primarily functions of upper limb, such as FMA-UE, WMFT and ARAT, patients with upper limb pain conditions such as rotator cuff tear, tennis elbow or trigger finger should be excluded in this study. The authors failed to list it into their exclusion criteria, which may result in a prominent confounding factor to the outcome measurement.
4. Line 483:The authors compared post-interventional scores instead of the amount of improvement. However, to compare the amount of improvement is more convincing, since the post-interventional scores were largely influenced by the pre-interventional scores.
5. The time after stroke were significantly different between the two groups, and the impact on the outcome measures should be discussed.
Author Response
Comment 1: Line 83: While the rationale of tenodesis principle is to perform wrist extension to assist hand grasping, the authors should explain why they designed WH-ENMS by assisting (1) wrist extension with the hand open and (2) wrist flexion with the hand closed, which is contradictory to the tenodesis principle.
Response: The tenodesis principle mainly adopts the passive tendon linkage of wrist extension related to finger flexion and wrist flexion associated with finger extension to facilitate patients with hand motor impairments poststroke (e.g., spasticity) achieving hand opening and grasping for daily tasks, when they cannot perform the full range of motions (ROMs) in the wrist and fingers as the unimpaired persons. The training tasks in this study, i.e., (1) wrist extension with the hand open and (2) wrist flexion with the hand closed, could help the patients achieve larger ROMs in both the fingers and the wrist joint with the assistances from the ENMS, comparing to those by the tenodesis principle, resulting in effective motor restoration as shown in the FMA and MAS improvements after the training.
Comment 2: Line 90: Why is WH-ENMS designed to detect EMG from extensor carpi ulnaris instead of extensor carpi radialis? Extensor carpi radialis is apparently a larger muscle and easier to locate.
Response: The WH-ENMS only used one pair of electrodes to capture the common EMG activity of the wrist and hand either in the extension phase, or in the flexion phase, with the purpose of easy operation in the self-help telerehabilitation. The selection of the extensor carpi ulnaris (ECU) muscle for EMG detection using the WH-ENMS system was based on the close anatomical proximity between the ECU and extensor digitorum (ED) muscles. Electrode pairs were placed in the common area of the motor point of the muscle bellies where ECU and ED muscles converge. This electrode configuration was successful for the device control [12] and feasible for the self-help operation [17].
Revisions: The above description has been added to the revised manuscript, Page 2, line 93 to page 3 line 97.
Comment 3: Line 155: Since the outcome measures of this study were primarily functions of upper limb, such as FMA-UE, WMFT and ARAT, patients with upper limb pain conditions such as rotator cuff tear, tennis elbow or trigger finger should be excluded in this study. The authors failed to list it into their exclusion criteria, which may result in a prominent confounding factor to the outcome measurement.
Revision: In the exclusion criteria, shoulder pain and poststroke central pain conditions have been added in page 4 lines 158-159.
Comment 4: Line 483:The authors compared post-interventional scores instead of the amount of improvement. However, to compare the amount of improvement is more convincing, since the post-interventional scores were largely influenced by the pre-interventional scores.
Response: Comparing the amount of change in the NUMs and MTD after the training by independent t-tests between the two groups, the lab group (mean = -20.0958; 95% confidence interval from -38.7219 to -1.4698) demonstrated a more significant decrease in the MTD (P = 0.014) than the clinic group (mean= 10.9202; 95% confidence interval from -5.8523 to 27.6926). No significant difference was observed in the NUMs (P = 0.742).
Revision: The results comparing the amount of changes in the MTD and NUMs have been added in Page12 Line 484-491.
Comment 5: The time after stroke were significantly different between the two groups, and the impact on the outcome measures should be discussed.
Revisions: In this work, the lab group was more chronic after a stroke than the clinic group. However, the lab group achieved better holistic motor improvements than the clinic group. It suggested that the time after stroke for the participants in this study was not a key determinant related to the rehabilitative outcomes. (Page 16, Line 613-616)
Reviewer 2 Report
The authors present a very thorough and well organized paper. All methods and results are presented in great detail. I have some methodological questions for the authors to address; Were the movement patterns repeated without variation during the training sessions? The text states 30 minutes for each task. While a 10-minute rest period was provided, did the authors account for mental fatigue or degradation of movement quality in the analysis? Also, what type of feedback was provided to the participants (e.g. prescriptive vs descriptive, and at what frequency and level of precision)? These are important motor learning consideration to increase intrinsic motivation. Moreover, was any variability built into the tasks or did the authors use only blocked practice parameters.
Results/Discussion: can the authors please provide more discussion on the differences between the USE and IMI inventories, specifically on the differences between the clinic and lab groups?
Minor edits:
Line 30: typo "asfter"
Author Response
Comment 1: Were the movement patterns repeated without variation during the training sessions? The text states 30 minutes for each task. While a 10-minute rest period was provided, did the authors account for mental fatigue or degradation of movement quality in the analysis.
Response: In the mandatory courses, the participants were taught on achieving the necessary quality in the repeated motions by using markers on the training table with desired distances in the horizontal and forward-reaching tasks (Page 5, Line 189). For the lab group, the participants were required to achieve the tasks with minimum body trunk compensations (Page 6, line 243-245). However, for the clinic group, the participants were allowed to use compensatory strategies from other body parts, e.g., trunk motions, to achieve the reaching targets (Page 7, Line 273-282). The motion quality of the participants in the bare-arm evaluation demonstrated that the lab group had better kinematic qualities after the training than the clinic group assessed by the parameters of NMUs and MTD (Figure 6 in the results). The mental fatigue and the movement quality monitoring in the training have not been measured in the current study, but will be investigated in our future work, which has been added in the discussion part (Page 17, Line 689-691).
Comment 2: What type of feedback was provided to the participants (e.g. prescriptive vs descriptive, and at what frequency and level of precision)? These are important motor learning consideration to increase intrinsic motivation.
Revisions: In the guided training sessions, onsite feedback by the operators was provided to enhance participants' performance and to support their transition to self-help training at home. They were primarily prescriptive in nature, focusing on specific instructions and corrective suggestions to ensure competence in performing the training exercises. (Page 5, line 205-208) In the self-help training sessions, operators’ feedback was tailored to solve technical issues encountered by participants, as well as to understand their overall training experiences. They were more descriptive in nature, aiming to provide explanations and guidance to help participants overcome challenges and improve their performance. The frequency and level of precision of the feedback varied depending on the specific needs and progress of each participant. (Page 5, Line 212-217)
Comment 3: Was any variability built into the tasks or did the authors use only blocked practice parameters.
Response: Individual parameters adopted in the WH-ENMS assistance were set for every participant in the first onsite session for successfully achieving the desired training tasks and were kept unchanged throughout the 20-session training. (Page 4, line 174-177) The variability in training tasks were adopted in the clinic group, comparing to the lab group. For example, the clinic group’s experimental operators were allowed to use different grasping objects in the training, such as a plastic apple, or a plastic cup, to simulate the objects with different shapes in daily living. An optional add-on limb task, i.e., a 30-minute forward task, was adopted by the operator for the participants in a training session for the clinic group, according to the center’s current practice in the upper limb rehabilitation. The participants were allowed to complete the training tasks with compensatory movements based on the pedagogy of the task-oriented rehabilitation, where compensation was also regarded as functional restoration once the task was achieved (Page 7, Line 273-282).
Comment 4: Results/Discussion: can the authors please provide more discussion on the differences between the USE and IMI inventories, specifically on the differences between the clinic and lab groups?
Revision:
Additional discussion on the USE and IME was added: ‘During the training in the clinic group, participants felt less supported and motivated to achieve motor restoration, which could be reflected by the lower scores of IMI compared with the lab group (Figure 7). As revealed by the USE results (Figure 7), participants in both groups reported holistic positive experiences on the usability of the WH-ENMS device ( > 50% normalized scores in all items). There was no difference between the groups on USE, although the clinic group suffered from fewer patient-operator interactions than the lab group. However, IMI results indicated that the clinic group showed less confidence or motivation in the training program than the lab group when professional support, or timely feedback, was reduced; and finally led to a significant group difference in perceiving the value/usefulness of the training program. Therefore, continuous support with timely feedback, whether onsite or remote, during the training could enhance the participants’ involvement and improve the overall outcome of the rehabilitation program.’ Page 17, Line 663-675
Minor Comments:
-Line 30: typo "asfter"
Revision: The related revision has been made in page 1, line 30.
Reviewer 3 Report
The manuscript describes a clinical study of rehabilitation-assisting robotic device for the restoration of hand movement after stroke.
Methods and results are clearly presented, conclusions of the paper are supported by the obtained results.
The overall scientific and technical level of the study is high.
The authors declare no conflict of interest, but at least one of the first authors is co-founder of Thecon Technology Hong Kong Ltd., which is the manufacturer of the ENMS device. The paper is based on the use of this device. I recommend to reflect this in the declaration of conflict of interest.
Overall level of English is good. There are typos, for instance:
Line 3 in the Introduction: “asfter”
Line 100: “Makers on the skin” – probably Markers
I recomend to carefully re-read the manuscript to eliminate typos.
Author Response
Comment 1: The authors declare no conflict of interest, but at least one of the first authors is co-founder of Thecon Technology Hong Kong Ltd., which is the manufacturer of the ENMS device. The paper is based on the use of this device. I recommend to reflect this in the declaration of conflict of interest.
Revision: Dr. Chingyi Nam is a co-founder of Thecon Technology Hong Kong Ltd. Thecon did not financially support the study. The ENMS devices in the study were purchased by the research team. Other authors have no conflicts of interest to declare. Page 18, line 727-729
Minor Comments:
- Line 30 in the Introduction: “asfter”
Revision: The related revision has been made in page 1, line 30.
-Line 100: “Makers on the skin” – probably Markers
Revision: The related revision has been made in page 3, line 101.
Round 2
Reviewer 1 Report
1. Although the authors included patients with MAS less than three as their study participants, some patients with mild spasticity still have unintentional EMG signals to provoke unwanted motion. How did the authors prevent such conditions in their study? For example, making wrist extension motion, which also produces stretching in wrist flexors, could create spasticity in the wrist flexors and unwanted EMG signals. How did the authors prevent this wrong EMG signal from being conveyed to WH-ENMS?
2. In the last revision, the authors excluded participants with shoulder pain. How about those with elbow and hand pain, such as tennis elbow or trigger finger? Could elbow and wrist pain affect the results of FMA-UE, WMFT, and ARAT? Did this exclusion reduce the number of participants? Furthermore, what about other neurologic conditions, such as Parkinsonism?
3. The authors recruited relatively few participants in this study without a preliminary sample size calculation, which could make the study lack statistical power.
4. In Table 2, no significant differences existed between the lab and clinic groups in age, gender, time since stroke, hemiplegic side, and type of stroke. How about the pre-interventional outcome measures? Were these measurements significantly different between the two groups?
Author Response
Thanks for the further comments and recommendations from the reviewers and the Editorial Board. Revisions in the manuscript were highlighted in red fonts with underlines in the manuscript. The revisions in the first round were kept in black fonts with underlines. The corresponding response to each comment is as follows:
Comment 1: Although the authors included patients with MAS less than three as their study participants, some patients with mild spasticity still have unintentional EMG signals to provoke unwanted motion. How did the authors prevent such conditions in their study? For example, making wrist extension motion, which also produces stretching in wrist flexors, could create spasticity in the wrist flexors and unwanted EMG signals. How did the authors prevent this wrong EMG signal from being conveyed to WH-ENMS?
Response: Poststroke spasticity is velocity-dependent involuntary muscle contracture mainly occurring in flexors [a], and usually could be triggered by fast joint rotation > 60o/s for stroke survivors with an MAS score of 3 [b,c,d,e]. The assistance from the WH-ENMS in the extension phase provided to the wrist and finger joints was associated with slow angular velocities of less than 15o/s [12]. Little excessive involuntary EMG was triggered in the wrist and finger flexors by the WH-ENMS assistance for the participants recruited in this work. The potential spasticity in the ENMS assistance also has been discussed in our previous work [12].
[a] Gerard E. Francisco and John R. McGuire, Poststroke Spasticity Management, Stroke, 2012; 43:3132-3136
[b] Deog Young Kim, Chang-il Park, Joong Son Chon, Suk Hoon Ohn, Tae Hoon Park, and In Keol Bang, Biomechanical Assessment with Electromyography of Post-Stroke Ankle Plantar Flexor Spasticity, Yonsei Med J. 2005 Aug 31; 46(4): 546–554.
[c] Raj T. S. Kumar, Anand D. Pandyan, Anil K. Sharma, Biomechanical measurement of post-stroke spasticity
Age and Ageing, Volume 35, Issue 4, 2006, Pages 371–375
[d] Condliffe EG, Clark DJ, Patten C. Reliability of elbow stretch reflex assessment in chronic post-stroke hemiparesis. Clin Neurophysiol 2005;116:1870–1878.
[e] Bhadane MY, Gao F, Francisco GE, et al. Correlation of resting elbow angle with spasticity in chronic stroke survivors. Front Neurol 2015;6:183.
Comment 2: In the last revision, the authors excluded participants with shoulder pain. How about those with elbow and hand pain, such as tennis elbow or trigger finger? Could elbow and wrist pain affect the results of FMA-UE, WMFT, and ARAT? Did this exclusion reduce the number of participants? Furthermore, what about other neurologic conditions, such as Parkinsonism?
Response: Subjects with chronic stroke are less suffered from tennis elbow (mainly caused by an overload of the elbow and repeated use of wrist and fingers) and trigger finger (mainly due to tendon dystopia), as the hemiplegic upper limb is much less used by survivors in the daily living after stroke leading to muscular atrophy. Compared to elbow and finger pain, shoulder pain is more commonly observed in chronic stroke, mainly because the muscle weakness at the shoulder causes the deformation of the joint, e.g., shoulder subluxation. Another potential cause of poststroke elbow or finger pain could be related to poststroke central pain, which has been added in the exclusion criteria in the last version. The participants recruited in this work did not suffer from any poststroke pain. Therefore, the revised exclusion criteria in the last version did not reduce the sample size of the study.
Revision: We did not recruit subjects with other neurological impairments besides stroke. This statement has been added to the updated exclusion criteria. (Page 4, line 159)
Comment 3: The authors recruited relatively few participants in this study without a preliminary sample size calculation, which could make the study lack statistical power.
Revisions: The limitation of the small sample size in the study has been added in the discussion part, ‘Although statistical significances were achieved in the intra- and inter-group comparisons of this work, large-scale and multi-center clinical trials with larger sample sizes will be carried out in future works to further validate the efficacy of the ENMS-assisted telerehabilitation poststroke.’ Page 17, Line 690-693
Comment 4: In Table 2, no significant differences existed between the lab and clinic groups in age, gender, time since stroke, hemiplegic side, and type of stroke. How about the pre-interventional outcome measures? Were these measurements significantly different between the two groups?
Response: There were no significant group differences in the pre-interventional outcome measures, as the results shown in Fig 4.